# Controllable Water Penetration through a Smart Brass Mesh Modified by Mercaptobenzoic Acid and Naphthalenethiol

Cong-Cong Luan [1], Yu-Ping Zhang [1,2,*], Cheng-Xing Cui [1], De-Liang Chen [3], Yuan Chen [3] and Meng-Jun Chen [3]

1    College of Chemistry, Zhengzhou University, Zhengzhou 450001, China
2    College of Chemistry and Materials Engineering, Hunan University of Arts and Science, Changde 415000, China
3    Changde Zhengyang Biotechnology Co., Ltd., Changde 415000, China
*    Correspondence: beijing2008zyp@163.com; Tel.: +86-199-36988587

**Abstract:** In this paper, a novel pH-responsive brass mesh modified by 3-mercaptobenzoic acid (MBA) and 2-naphthalenethiol (NPT) was demonstrated via a facile chemical etching method followed by surface modification. The smart wettability was dependent on the assembled MBA and NPT with suitable thiol proportions. The on–off control of water penetrating intelligently into the nanostructured brass mesh substrate was carried out by the pH change in the outside environment. The brass mesh modified with $X_{NPT} = 0.4$ (mole fraction of NBT in the mixed solution) exhibited the strongest pH responsivity from superhydrophobicity to superhydrophilicity. Furthermore, the resulted Janus membrane (JM) fabricated by the integration of a smart brass mesh and hydrophobic Ni foam could be used as a water diode in air and liquid systems. Unidirectional penetration for the water droplet was realized by the resulting smart JM with a hydrophobic upper layer and a pH-responsive layer below.

**Keywords:** smart material; brass mesh; 3-mercaptobenzoic acid; 2-naphthalenethiol; Janus membrane

## 1. Introduction

Stimuli-responsive materials have been widely studied for their different interesting and practical applications [1,2]. Such materials are fabricated through the polymerization of some specific monomers, or via the physical or chemical attachment of stimuli-responsive compounds on the selected substrates. Depending on the stimuli-responsive molecules on the substrate surface, it is possible to design different smart surfaces with a wide range of responsivities toward light, temperature, pH, force, etc. [3–6].

Liu et al. fabricated a kind of porous polysulfone microcapsule by mixed solvent volatilization, in which a coating of stearic acid acted as a pH-responsive smart microcapsule slow-release filler. The resultant microcapsules exhibited a typical pH-triggering performance in an alkaline environment [4]. Yang et al. developed a spray-coating method to prepare a flexible surface, whose reversible switch was realized between hydrophobic and hydrophilic states under UV/vis irradiation. The resulting intelligent material was successfully used for "oil-removing" and "water-removing" by varying the lighting mode. Moreover, it could repeatedly withstand mechanical deformation during multiple practical applications [5]. Štular and et al. synthesized two kinds of hydrogels, namely poly(N-isopropylacrylamide) and chitosan, with an average particle size of 405 nm and 76 nm, respectively. Both hydrogels were used to produce poly(lactic acid) fabric and, the temperature and pH responsiveness of the modified fabrics were investigated based on the moisture content, water uptake, and water vapor transition rate [6]. At present, smart materials with pH responsiveness are attracting widespread concern due to their versatile applicability in water transformation systems [7–9]. Such a modification enables the material to respond smartly to variation in the pH of an aqueous solution. Guo's group demonstrated a novel pH-responsive liquid marble covered by thiol-modified copper powders through a facile

one-step self-assembly method [10]. The protonation and deprotonation processes of copper powder modified by $HS(CH_2)_9CH_3$ and $HS(CH_2)_{10}COOH$ can be tailored via the pH change from the external environment, and the controllable and switchable wettability of copper powder was used to maintain or break liquid marble. Cheng et al. reported a novel strategy to control water permeation on a copper mesh modified by $HS(CH_2)_9CH_3$ and $HS(CH_2)_{10}COOH$ [11]. Shi's group described a combined approach to prepare a smart material including the initial electroless deposition of gold and subsequent immersion in a mixed solution of $HS(CH_2)_{11}CH_3$, $HS(CH_2)_{10}COOH$, and $HS(CH_2)_{11}NH_2$. The resultant surface underwent a reversible transformation with the pH change in the external liquid droplets [12]. Liu et al. fabricated a novel pH-responsive smart device by electroless silver deposition followed by surface modification with a mixed thiol solution of $HS(CH_2)_{11}CH_3$ and $HS(CH_2)_{10}COOH$. It was successfully applied for continuous separations of oil/water mixtures using the as-prepared copper foams, and the surface wettability was tailored reversibly between superhydrophobicty and hydrophilicity by varying the pH value of the aqueous solution [13]. Wang et al. reported a pH-responsive Janus membrane (JM) modified by $HS(CH_2)_{10}COOH$, which could be used for responsive gating and the unidirectional transformation of water droplets [14]. Zhang et al. prepared pH-responsive smart non-woven fabrics with reversible transformations via the in situ grafting of Ag nanoparticles through redox between $AgNO_3$ and ascorbic acid. After modification with a mixed thiol solution, including $HS(CH_2)_{10}CH_3$, $HS(CH_2)_{10}COOH$ and $HS(CH_2)_{11}OH$, oil/water separation was successfully carried out using the fabricated fabrics [15].

Herein, in order to extend the immobilized stimuli-responsive unit on the metal substrates, we firstly attempted to self-assemble two kinds of novel pH-responsive molecules of 3-mercaptobenzoic acid ($HS-C_6H_4-COOH$) and 2-naphthalenethiol ($HS-C_{10}H_7$) on an etched brass mesh. The pH-responsive SAMis formed by chemisorption of both of the novel thiol-molecules onto the substrate, resulting in different surface wettability in responses to the pH change. Moreover, in situ grafting, depositing, or spraying of Ag or Au nanoparticles was avoided before the modification of the pH-responsive molecules compared with previous studies [10–15].

## 2. Results and Discussion

### 2.1. Characterization of the Smart Brass Mesh

Figure 1(a1) shows the morphology of the pristine brass mesh. The diameter of one brass wire and the size of one square pore were about 40 μm and 58 μm, respectively. The brass skeleton exhibited a smooth surface morphology, and no thorns were observed as shown in the enlarged SEM images illustrated in Figure 1(a2). On the contrary, the brass skeleton surface was uniformly covered by a large number of thorns on a micro/nano-scale after being chemically etched as shown in Figure 1(b1) and in the high-magnification image of Figure 1(b2). The pine-needle-shaped structures on its surface had a length of a few micrometers and each pine needle consisted of many tiny branches of a few microns or a few hundred nanometers. Figure 1c further indicates the structure of the coating layer with thin self-assembled monolayers of thiol-compounds after modification. The surface of the brass skeleton was tightly covered by some pine-needle-shaped structures with a thin layer of film. These hierarchical structures with micro-/nanoscale dimensions contributed to the maximized change in the surface wettability between the superhydrophobic and superhydrophilic states.

The EDS in Figure 1d shows the elements of C, O, Cu, Zn, and S with different contents on the modified brass mesh treated by the mixed thiols. XPS was used to observe the chemical composition of the thin film of the etched surface. As shown in Figure 1f, the XPS spectrum displays the presence of C, O, Cu, Zn, and S on the as-prepared substrates.

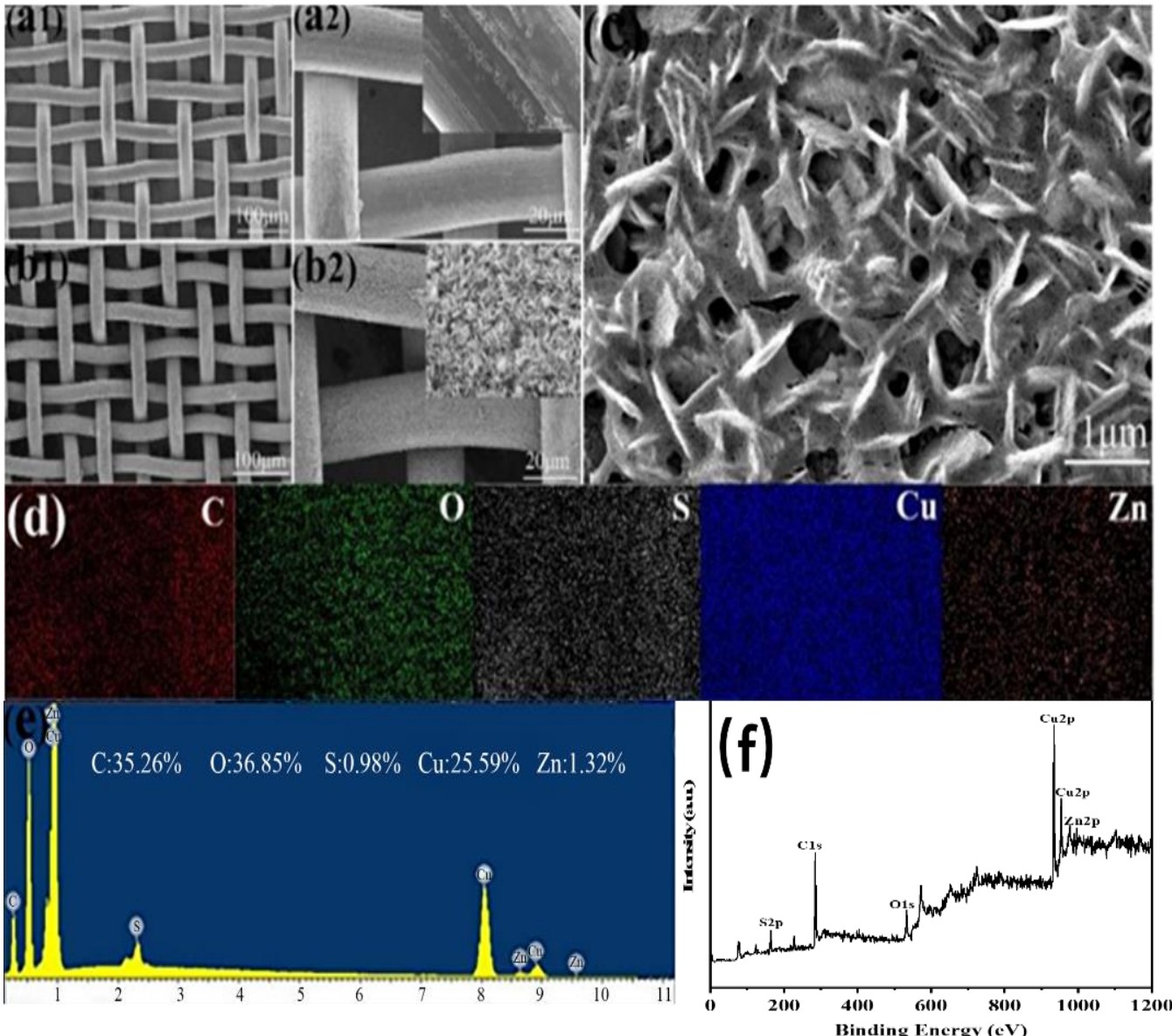

**Figure 1.** Characterization of the pristine, etched, and modified brass meshes by SEM, EDS, and XPS. (**a1**) The morphology of the pristine brass mesh;(**a2**) The enlarged SEM images for the morphology of the pristine brass mesh; (**b1**) The morphology of the chemically etched brass mesh; (**b2**) The enlarged SEM images for the morphology of etched brass mesh; (**c**)The morphology of the modified brass mesh with a thin SAM; (**d**) EDS mapping image of the modified brass mesh; (**e**) EDS analysis of the modified brass mesh; (**f**) XPS spectrum of the modified brass mesh.

## 2.2. Optimazation of Surface Modification

We observed that the molar ratio of both molecules in the modified solution played an important role for the surface wettability of the brass mesh. A series of brass meshes were immersed in a mixed solution of MBA and NPT with different molar ratios, and the wetting properties of the prepared films were investigated using CA measurements. As shown in Figure 2a, when $X_{NPT} = 0.5$, the rough film was superhydrophobic for neutral water, while it was hydrophilic for basic water. When $X_{NPT} = 0.2$, the rough film was superhydrophilic for basic water, while for a neutral water droplet, the contact angle only reached about 133°. When further decreasing or increasing the value of $X_{NPT}$, the obtained films could not create a significant change in the surface wettability. If only MBA was immersed for modification, the resulting surface was superhydrophilic to both neutral and basic droplets.

Meanwhile, the resulting surface only modified by NPT was superhydrophobic to both neutral and basic droplets. Obviously, the brass mesh immersed with $X_{NPT} = 0.4$ (mole fraction of NBT in the mixed solution) exhibited the best pH responsivity from being superhydrophobic to superhydrophilic. In comparison, on the smooth brass mesh modified with mixed thiol ($X_{NPT} = 0.4$), the contact angles (CAs) for the neutral water droplets (10 μL, pH = 7) and the basic water droplets (10 μL, pH = 12) were only about 103° and 30° due to the absence of chemical etching, respectively, while on the rough brass mesh substrate, an enhanced effect was obtained, with the largest transformation for the surface wettability. Although the smooth mesh film had a similar wettability transition between hydrophilicity and hydrophobicity, it was unfit for unidirectional water permeation. Due to the lower WCAs for the neutral droplets compared to the rough one, such a smooth mesh film was not highly hydrophobic enough to avoid water permeation under a small water pressure (even a water droplet) or vibration. Therefore, the modified rough brass mesh was selected for the further study of controllable water permeation.

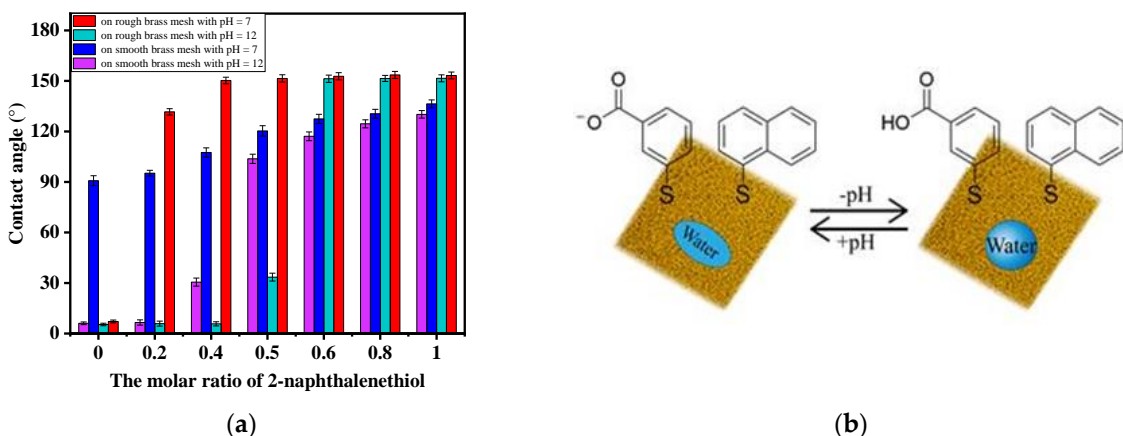

**Figure 2.** Dependence of the contact angles on $X_{NPT}$ (**a**), the mechanism of pH-responsive property (**b**).

During self-assembly, intermolecular forces, such as coordination interactions, van der Waals' forces, hydrogen bonds, and solvophobic effects, usually play a dominant role for the connection of smart molecular units in a reversible way in self-assembled structures. From the above investigation, it could be seen that the controllable water permeation was dependent on the pH-responsive film, and such a responsive ability was attributed to the following two reasons. One was the protonation and deprotonation of the surface carboxylic acid groups in the different pH environments, respectively. The other was the nanostructures on the brass mesh, which enhanced the wettability transformation. The relative mechanism of the pH-responsive properties is illustrated in Figure 2b. In general, the carboxyl group of the NPT molecules is deprotonated to -COO⁻ under basic conditions, resulting in the superhydrophilicity of the brass mesh. On the contrary, the deprotonated -COO⁻ is protonated back to -COOH under acidic conditions, resulting in the hydrophobicity of the mesh. Water with different pH values can be used to trigger the switch in surface wettability.

In order to achieve the desired transformation between the superhydrophobic and superhydrophilic states, the mole ratio of $HS-C_6H_4-COOH$ (MBA) and $HS-C_{10}H_7$ (NPT) was thus set to 4/6. The as-prepared mesh films were fabricated for the investigation of water-controllable penetration. A rough brass mesh folded into a box was placed on the top of a small beaker and some neutral water was poured into it carefully. Figure 3a shows that the neutral water was blocked and held in the box. When alkalescent water was poured into the box, it penetrated the mesh and dropped down into the beaker (Figure 3b). It appeared that unidirectional water transportation using the smart brass mesh could be realized by simply controlling the water pH value.

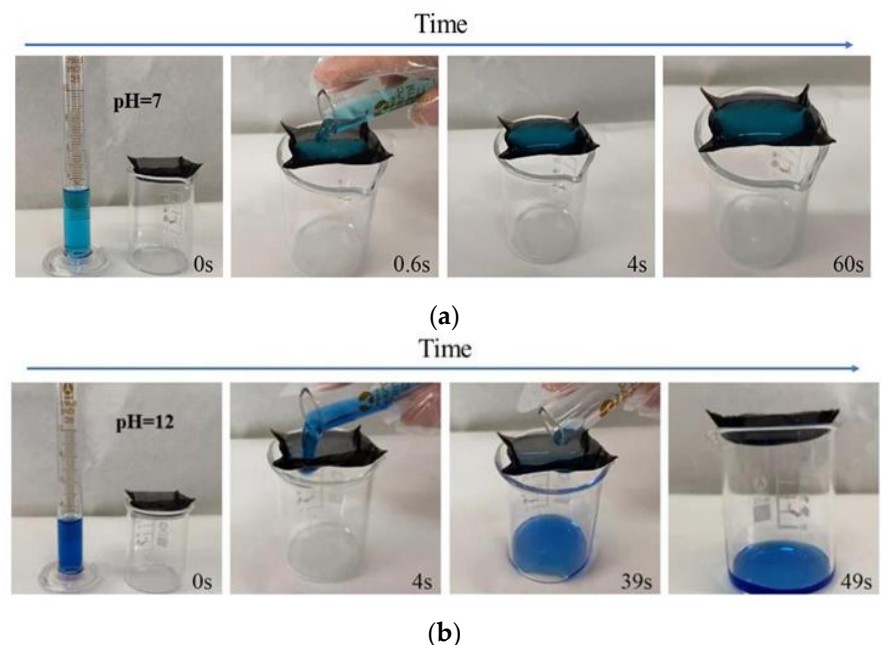

**Figure 3.** Illustrated images of the unidirectional water transportation. Acidic and neutral water were blocked by the rough mesh film (**a**); unidirectional penetration occurred when the pH of the water was increased to 12 (**b**). Water was dyed blue by methylthionine chloride.

*2.3. Application of Controllable Water Permeation*

In order to investigate the controllability of water permeation for the as-prepared mesh films, some droplets with different pH values were selected for further experiment. As shown in Figure S1, the surface was superhydrophobic when the pH value of the aqueous solution was less than 7. Yet, the WCA sharply decreased when the pH value was more than 11. When the pH value was greater than 12, the water could rapidly penetrate the void of the brass mesh due to its superhydrophilic property. Furthermore, after being cleaned with ultrapure water and dried by nitrogen blowing, the brass mesh film recovered its superhydrophobic performance for cyclic utilization more than ten times, demonstrating that the resultant film was stable, and the simple device was potentially suitable for practical applications (see Figure S2).

Another approach to control water penetration was illustrated using a Janus membrane (JM) [16–18]. Herein, a JM was constructed by the integration of the smart brass mesh and a flexible pristine Ni foam by a tablet press under a pressure of 10 MPa. As shown in Figure S3, the framework of the hydrophobic nickel foam was flat and smooth, and on the surface of smart brass mesh, a layer full of countless hair-like nanoneedles embedded in the thin film was observed. The resulting JM was integrated by a front layer of Ni foam and a back layer of smart brass mesh, and its behavior could transform from homogeneous hydrophobicity to asymmetric wettability with the pH change in the water droplets. For example, unidirectional penetration occurred for a water droplet with a pH of 12 in the liquid–air system. When a basic water droplet (pH = 12) was seated on the front layer of the resulting "JM", it slowly penetrated the thin and hydrophobic Ni foam. When it came into contact with the pH-responsive layer below, the contact point of the pH-responsive layer with the basic droplet changed from hydrophobic to highly hydrophilic, and thus unidirectional penetration around this narrow area occurred, demonstrating that this area had Janus characteristics in the air–liquid system. As shown in Figure S4, only basic water droplets penetrated spontaneously from the hydrophobic to the hydrophilic side (the positive direction), whereas when the mesh film was turned over (the reverse direction), the water droplets with different pH (pH = 2 and 7) values were blocked, and the water droplets (pH = 12) spread on the smart film. A similar behavior of water droplet unidirectional transportation was observed in the liquid–liquid system, as shown in Figure S5.

The controllable penetration of water droplets was also realized using the resulting JM in an oil–water system, which is illustrated in Figure 4. The transportation process was closely related to the pH value of the aqueous solution below. When the acidity of the aqueous solution below was adjusted to pH = 2, the whole membrane was evenly hydrophobic, which was composed of the front Ni foam layer and the back layer of brass mesh (pH-responsive). The high hydrophobic force from both layers effectively prevented the water droplets from permeating. Therefore, the dyed-blue water droplets (containing 0.1 mM $CuSO_4$) could not penetrate the thick hydrophobic membrane, regardless of whether the "JM" was positively or reversely aligned at the oil/water interface (see Figure 4a,b). Furthermore, when the back layer (pH-responsive layer) contacted the basic aqueous solution (pH = 12), it transformed to be highly hydrophilic. With the gradual permeation of the blue water droplets, blue floc precipitation of $Cu(OH)_2$ was apparently created due to the reaction between $Cu^{2+}$ and $OH^-$ in the bottom solution (see Figure 4c). When the pH-responsive JM was reversely aligned at the oil–water interface, blockage of the blue water droplets occurred (see Figure 4d).

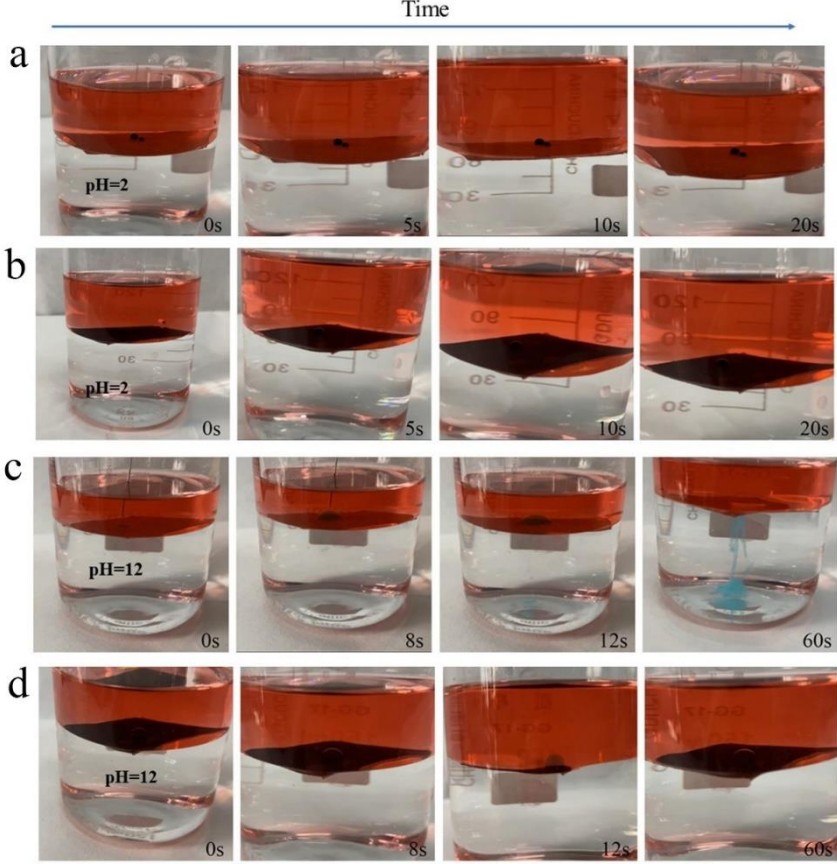

**Figure 4.** Directional penetration of water droplets across the resulting "JM" in oil–water systems. (**a**,**b**) Positively and reversely aligned "JM" floating on the water solution (pH = 2) with a water droplet seated on the integrated membrane with both hydrophobic sides. (**c**) Positively aligned "JM" with a water droplet seated on the hydrophobic side creating a larger driving force for penetration due to the superhydrophilic bottom layer. (**d**) Reversely aligned "JM" with a water droplet seated on the smart side was subject to spreading without penetration.

## 3. Experimental Section

### 3.1. Reagent and Material

3-Mercaptobenzoic acid (MBA) and 2-naphthalenethiol (NPT) were bought from Shanghai Aladdin Biochemical Co., Ltd. (Shanghai, China). Oil red (Sudan III), methylene blue, $(NH4)_2S_2O_8$, NaOH, HCl, and n-hexane were purchased from Tianjing Fine Chemical

Co. (Tianjing, China). Brass mesh with an aperture of 300 was purchased from Dongguan Shijia Metal Materials Co., Ltd. (Guangdong, China).

### 3.2. Preparation of Smart Brass Meshes

The pristine brass meshes were consequently cleaned for 5 min with 1% HCl solution and deionized (DI) water in order to remove the native oxide on the surface. Then, they were immersed in an aqueous solution of 2.5 M NaOH and 0.1 M $(NH_4)_2S_2O_8$ for more than 30 min [19,20]. After being etched, the resulting brass meshes were washed with DI water followed by $N_2$ flow-drying. The obtained brass meshes became rough and superhydrophilic due to the full form of hair-like $Cu(OH)_2$ nanowires on their surfaces. After immersion in the mixed solutions (5 mM) of 3-mercaptobenzoic acid (MBA) and 2-naphthalenethiol (NPT) overnight, the surface energy was reduced because of the grafted benzene ring and naphthalene ring.

### 3.3. Measurement of Wetting Performance

The water contact angles (WCAs) were measured using a contact angle goniometer (TST-200, Shen Zhen Testing Equipment Co. LTD., Shen Zhen, China). The WCAs were measured by dropping $10 \pm 0.5$ μL droplets on the substrate using a micrometer syringe. At least six measurements were performed on each brass mesh attached to a glass slide.

### 3.4. Characterization of the Brass Mesh

The pristine and modified brass meshes were characterized using a SEM 98 (Hitachi Su5000, Hitachi, Japan) and EDS (Oxford Instruments Ultim Max, Oxford, UK). The surface elements and composition of the coatings were measured by XPS (Thermo Fisher, Waltham, MA, USA).

### 4. Conclusions

In this study, two controllable methods of water permeation were demonstrated. One was carried out through an etched brass mesh modified by two kinds of responsive molecules. The doubly transforming surface modified with the mixed SAMs of NPT and MBA exhibited pH responsiveness from superhydrophobic to superhydrophilic. The other was conducted by constructing a JM with hierarchical micro/nanostructures and pH-responsive properties, which could be used as a water diode for unidirectional droplet transportation. This work provides an alternative strategy to explore this novel smart material and its JM for controllable water permeation.

**Supplementary Materials:** The following supporting information can be downloaded at: https://www.mdpi.com/article/10.3390/coatings12111729/s1, Figure S1: The water contact angle (WCA) curve with the pH change; Figure S2: Reversible transition between the superhydrophobicity and superhydrophilicity on the rough copper mesh film (prepared with $X_{COOH} = 0.6$) could be repeated by alternately changing the water pH; Figure S3. SEM of the integrated membrane of hydrophobic Ni foam and smart brass mesh; Figure S4: Directional water droplet penetration across the integrated membrane in the air—-water system. (a) The integrated membrane only allowed penetration of alkaline water droplets (pH = 13) when it was positively aligned and (b) prevented all water droplet (pH = 2, 7, and 13) penetration when reversely aligned; Figure S5: An "on–off" control for water droplets with different pH (pH = 2, 7 and 13) in the oil–water system. (a) Liquid droplets were blocked when the integrated membrane was positively aligned at the interface; (b) liquid droplets were blocked when the membrane was negatively aligned at the interface. Light oil (n-hexane) was dyed red by Sudan III.

**Author Contributions:** Conceptualization, C.-X.C.; Data curation, Y.C. and M.-J.C.; Software, D.-L.C.; Writing – original draft, C.-C.L.; Writing – review & editing, Y.-P.Z. All authors have read and agreed to the published version of the manuscript.

**Funding:** Financial support was provided from the National Nature Science Foundation of China (No. 22074029); the Scientific Innovation Team in Henan Province (No. C20150020); and the Start-up Project for High Level Talents in HUAS.

**Institutional Review Board Statement:** Not applicable.

**Informed Consent Statement:** Not applicable.

**Data Availability Statement:** Data are contained within the article.

**Conflicts of Interest:** The authors declare no conflict of interest.

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
