# Peer review of "Controllable Water Penetration through a Smart Brass Mesh Modified by Mercaptobenzoic Acid and Naphthalenethiol"

_coatings, doi:10.3390/coatings12111729_

Round 1

Reviewer 1 Report

The research paper entitled "Controllable Water Permeation through a Smart Brass Mesh Modified by Mercaptobenzoic Acid and Naphthalenethiol" is interesting that can be considered for publication in the journal Coatings.

However, major points should be addressed as bellows:

- Abstract needs more quatitative information

- Introduction of polymer application in coatings is too less.

The authors should refer some papers:

Progress in Organic Coatings 158, 106361; Langmuir 36 (43), 13001-13011; Journal of Molecular Liquids 309, 113150; Journal of Molecular Liquids 339, 116754; Progress in Organic Coatings 174, 107249

- Resolution of Figure 1 needs improve 

- Discussion on surface modification is too less.

- Electrostatic interactions are not clear.

Author Response

  1. Abstract needs more qualitative information

Answer: we have added the key qualitative information in the abstract as follow: The brass mesh modified with XNPT = 0.4 (mole fraction of NBT in the mixed solution) exhibited the best pH-responsivity from the superhydrophobicity to the superhydrophilicity.

2.-Introduction of polymer application in coatings is too less. The authors should refer some papers: Progress in Organic Coatings 158, 106361; Langmuir 36 (43), 13001-13011; Journal of Molecular Liquids 309, 113150; Journal of Molecular Liquids 339, 116754; Progress in Organic Coatings 174, 107249
Answer: Many thanks for the suggestion from the reviewer! Herein, we have added some polymer application in smart coatings in the introduction as follows:

Liu and et al fabricated a kind of porous polysulfone microcapsules by mixed solvent volatilization, in which a coating of stearic acid acted as a pH-responsive smart microcapsule slow-release filler. The resultant microcapsules exhibited typical pH-triggering performance in an alkaline environment. (See Ref.4)

Yang and et al developed a spray-coating method to prepare a flexible surface, which reversible switch was realized between hydrophobic and hydrophilic states under UV/vis irradiation. The intelligent material was successfully used for “oil-removing” and “water-removing” by alternate lighting mode. Moreover, and it could withstand mechanical deformation repeatedly during multiple practical applications. (See Ref.5)

Štular and et al synthesized two kinds of hydrogels based on poly(N-isopropylacrylamide) and chitosan with an average particle size of 405 nm and 76 nm, respectively. Both hydrogels were applied to poly(lactic acid) fabric and the temperature and pH responsiveness of the functionalised fabric were investigated based on the water uptake , moisture content and water vapour transition rate. (See Ref.6)

3- Resolution of Figure 1 needs improve 

Answer: we have redrawn Fig.1, now  the resolution of Fig.1 is improved.

4- Discussion on surface modification is too less.

Answer: we have added some discussion on surface modification.

5- Electrostatic interactions are not clear.

 Answer: Many thanks for the reviewer’s comment. Herein, protonation and deprotonation effects under different pH conditions are the main reason for smart responsivity, similar to previous studies using mixed thiol compounds.  We have revised the mechanism in our revised manuscript as follows: During the self-assembly, the intermolecular forces such as coordination interactions, van der Waals' forces, hydrogen bonds, and solvophobic effects usually plays a dominant role for the connection of the smart molecular units in a reversible, controllable way in the self-assembled structures.

Reviewer 2 Report

- Manuscript has some typos, revise carefully and correct it

- Introduction is difficult to understand, try to improve it

- Last part of the introduction, include sentence “the aim of …” or “the objective of this research” etc.

- Figure 1 were difficult to see caption for Figures (a1), (b1), (c)

- Figure 1 (f) is no possible to see the information, correct it

- Figure 2 (a) was no possible to see the information for different samples (blue, red, etc.)

- All Figures need to improve at minimal 300 dpi, because in the present form Figures have very poor resolution

- Manuscript has some interesting result but doesn’t have discussion, include discussion for all Figures

- 3. Experimental. Include information about materials (chemical compounds, etc.) and purification process or include sentence “all materials used as received”

- Conclusion needs to improve

- Manuscript has 2 references from 2021, include references from 2022

Author Response

1- Manuscript has some typos, revise carefully and correct it

Answer: we have corrected some typos in our manuscript.

2- Introduction is difficult to understand, try to improve it

Answer: we have revised the introduction in order to enhance the clear understanding.

3.- Last part of the introduction, include sentence “the aim of …” or “the objective of this research” etc.

Answer: we have added some sentences to improve the understanding about the aim of our work.

4.- Figure 1 were difficult to see caption for Figures (a1), (b1), (c)

Answer: we have corrected Figure 1 in order to improve the resolution.

5.- Figure 1 (f) is no possible to see the information, correct it

Answer: we have redrawn Figure 1, now Figure 1(f) is clearer.

6.- Figure 2 (a) was no possible to see the information for different samples (blue, red, etc.)

Answer: we have redrawn Fig.2(a), now the information for different samples is clearly observed.

7.- All Figures need to improve at minimal 300 dpi, because in the present form Figures have very poor resolution

Answer: we have improved all figures with poor resolution

8.- Manuscript has some interesting result but doesn’t have discussion, include discussion for all Figures

Answer: we have added some discussion in our revised manuscript, which is involved in all figures now.

Self-assembly is an important fabrication method in which components spontaneously form ordered aggregates by noncovalent interactions, such as electrostatic interactions, hydrogen bonds, van der Waals' forces, coordination interactions and solvophobic effects. It is considered a general approach for fabricating smart surfaces because the intermolecular forces connect the molecular building blocks in a reversible, controllable way in the self-assembled structures.

  1. Experimental. Include information about materials (chemical compounds, etc.) and purification process or include sentence “all materials used as received”

Answer: we have added some information about chemical compounds as follows:

3-mercaptobenzoic acid (MBA) and 2-naphthalenethiol (NPT) were bought from Shanghai Aladdin Biochemical Co., Ltd. (Shanghai, China). Oil red (Sudan III), methylene blue, (NH4)2S2O8, NaOH, HCl, and hexane were purchased from Tianjing Fine Chemical Co., China. Brass mesh with an aperture of 300 mesh was purchased from Dongguan Shijia Metal Materials Co., Ltd. Fish feed was purchased from a local supermarket.

10.- Conclusion needs to improve

Answer: we have added some sentences in our conclusion.

11- Manuscript has 2 references from 2021, include references from 2022

Answer: we have added some new references from 2021 and 2022. Please see References 5-6 in our revised manuscript.

Round 2

Reviewer 1 Report

The revised manuscript is significantly improved by the authors.

The paper can be accepted in current form.